# Unraveling the Enigma of Double Descent: An In-depth Analysis through the Lens of Learned Feature Space

**Yufei Gu**[1,3]*, **Xiaoqing Zheng**[1,2], **and Tomaso Aste**[3]
[1]School of Computer Science, Fudan University
[2]Shanghai Key Laboratory of Intelligent Information Processing
[3]Department of Computer Science, University College London
`{yufei.gu.20, t.aste}@ucl.ac.uk, zhengxq@fudan.edu.cn`

## Abstract

Double descent presents a counter-intuitive aspect within the machine learning domain, and researchers have observed its manifestation in various models and tasks. While some theoretical explanations have been proposed for this phenomenon in specific contexts, an accepted theory for its occurring mechanism in deep learning remains yet to be established. In this study, we revisit the phenomenon of double descent and demonstrate that the presence of noisy data strongly influences its occurrence. By comprehensively analysing the feature space of learned representations, we unveil that double descent arises in imperfect models trained with noisy data. We argue that while small and intermediate models before the interpolation threshold follow the traditional bias-variance trade-off, over-parameterized models interpolate noisy samples among robust data thus acquiring the capability to separate the information from the noise. The source code is available at `https://github.com/Yufei-Gu-451/double_descent_inference.git`.

## 1 Introduction

As a well-known property of machine learning, the bias-variance trade-off suggested that the variance of the parameter estimated across samples can be reduced by increasing the bias in the estimated parameters (Geman et al., 1992). When a model is under-parameterized, there exists a combination of high bias and low variance. Under this circumstance, the model becomes trapped in under-fitting, signifying its inability to grasp the fundamental underlying structures present in the data. When a model is over-parameterized, there exists a combination of low bias and high variance. This is usually due to the model over-fitting noisy and unrepresentative data in the training set. Traditional statistical intuition suggests a difficulty in reducing bias and variance simultaneously and advises finding a 'sweet spot' between underfitting and overfitting. This conjugate property prevents machine learning estimators from generalizing well beyond their dataset (Hastie et al., 2009). However, modern machine learning practices have achieved remarkable success with the training of highly parameterized sophisticated models that can precisely fit the training data ('interpolate' the training set) while attaining impressive performance on unseen test data.

To reconcile this contradiction between the classical bias-variance trade-off theory and the modern practice, Belkin et al. (2019) presented a unified generalization performance curve called 'Double Descent'. The novel theory posits that, with an increase in model width, the test error initially follows the conventional 'U-shaped curve', reaching an interpolation threshold where the model precisely fits the training data; Beyond this point, the error begins to decrease, challenging the traditional understanding and presenting a new perspective on model behavior. Similar behaviour was previously observed in Vallet et al. (1989); Opper (1995; 2001), but it had been largely overlooked until the double descent concept was formally raised.

---

*This work was done during the author's internship at Fudan University.

The double descent phenomenon is prevalent across various machine learning models. In the context of linear models, a series of empirical and theoretical studies have emerged to analyze this phenomenon (Bartlett et al., 2020; Loog et al., 2020; Belkin et al., 2020; Deng et al., 2022; Hastie et al., 2022). In deep learning, many researchers have offered additional evidence supporting the presence and prevalence of the double descent phenomenon across a broader range of deep learning models, including CNNs, ResNets, and Transformers, in diverse experimental setups (Nakkiran et al., 2021; Chang et al., 2021; Liu et al., 2022; Gamba et al., 2022). Nakkiran et al. (2021) specifically concluded that this phenomenon is most pronounced when dealing with label noise in the training set, while can manifest even without label noise for the first time. Despite the widely accepted empirical observation of the double descent phenomenon within the domain of deep learning, comprehending the underlying mechanism of this occurrence in deep neural networks continues to pose a significant question.

It is well-known that deep neural networks accomplish end-to-end learning frameworks through integrating feature learning with classifier training (Zhou, 2021). A neural network tailored for classification tasks can be divided into two components. The initial segment is dedicated to a feature space transformation process by selecting information pertinent to the output while discarding information associated with the input. Specifically, the conversion of the original feature space represented by the input layer to the final feature space represented by the final representation layer is referred to as the learned feature space, which is shaped by the acquired representations within fully-trained neural networks. The subsequent stage typically entails constructing a classifier based on this learned feature space, intending to complete machine-learning tasks.

In this paper, we highlight the crucial role that noisy samples play in shaping the behaviour of deep learning models through a comprehensive exploration of the interplay among model width, noisy training data, and interpretation of the learned feature space. We demonstrated that while small and intermediate models follow the traditional bias-variance trade-off, over-parameterized models which passed the interpolation threshold, interpolate more correct training data around noisy ones of matching classes. This phenomenon demonstrates a correlation with the double descent phenomenon, and we propose that this is attributed to models acquiring the ability to effectively 'isolate' noise from the information in the training dataset. We posit that this phenomenon has not been previously documented and could serve to elucidate the internal mechanism of double descent.

The main contributions of this study are summarized as follows. We proposed that there exists a robust correlation between the neural network's interpolation strategy of noisy data and the double descent phenomenon. Utilizing the $k$-nearest neighbour algorithm, we indicate that this phenomenon results from broader architectures genuinely interpolating noisy samples among correct samples of the same original class, showcasing an enhanced capability to isolate noise information. Through conducting experiments on various neural architectures trained on MNIST and CIFAR-10, we demonstrated that the label prediction accuracy of noisy samples strongly aligns with the test performance on unseen data. We believe our investigation can shed light on the mechanism by which over-parameterization enhances generalization.

## 2 RELATED WORK

The initial proposal of the double descent phenomenon as a general concept was made by Belkin et al. (2019) for increasing model size (model-wise). However, Nakkiran et al. (2021) have also shown a similar trend w.r.t. size of the training dataset (sample-wise) and training time (epoch-wise). The research further reports observation of all forms of double descent most strongly in settings with label noise in the train set. While the double descent phenomenon is widely discussed in these alternative perspectives, we focus our attention on the original model-wise phenomenon in this work, briefed as 'double descent' in the subsequent discussion.

Double Descent has been shown as a robust phenomenon that occurs over diverse tasks, architectures, and optimization techniques. Subsequently, a substantial volume of research has been conducted to investigate the double descent phenomenon regarding its mechanism. A stream of theoretical literature has emerged, delving into the establishment of generalization bounds and conducting asymptotic analyses, particularly focused on linear functions (Advani et al., 2020; Belkin et al., 2020; Bartlett et al., 2020; Muthukumar et al., 2021; Mei & Montanari, 2022; Hastie et al., 2022). Hastie et al. (2022) highlight that when the linear regression model is misspecified, the best

generalization performance can occur in the over-parameterized regime. Mei & Montanari (2022) further concluded that the occurrence of the double descent phenomenon can be attributed to model misspecification, which arises when there is a mismatch between the model structure and the model family. While people believe the same intuition extends to deep learning as well, nevertheless, the underlying mechanism still eludes a comprehensive understanding of 'deep' double descent.

The prevailing literature that explores the double descent phenomenon in deep neural networks learning provides a range of explanations: bias-variance decomposition (Yang et al., 2020); samples to parameters ratio (Belkin et al., 2020; Nakkiran et al., 2021); decision boundaries (Somepalli et al., 2022); and the sharpless of interpolation (Gamba et al., 2022). While offering an extensive array of experimental findings that replicate the double descent phenomenon, Nakkiran et al. (2021) concluded that the emergence of double descent exhibits a robust correlation with the inclusion of noisy labels in the training dataset. Somepalli et al. (2022) further studied the decision boundaries of random data points and established a relationship between the fragmentation of class regions and the double descent phenomenon. Recently, Chaudhary et al. (2023) revisited the phenomenon and proposed that double descent may not inherently characterize deep learning.

Beyond the thorough examinations of double descent, an emerging theory of over-parameterized machine learning (TOPML) is growing, seeking to elucidate the second 'descent' in the double descent phenomenon (Dar et al., 2021). The inquiry posed by Zhou (2021) revolves around the puzzling aspect of why over-parameterized models do not succumb to overfitting and focus on the feature space transformation aspect. Chang et al. (2021) analytically identify regimes when training over-parameterized networks and then pruning is preferable to training small models directly. Teague (2022) intriguingly explores the concept of implicit regularization arising from over-parameterization and draws a connection between the generalization capacity and the geometric properties of volumes within hyperspaces. The work by Liu et al. (2022) introduces a robust training method for over-parameterized deep networks in classification tasks, specifically when training labels are corrupted. Although our research shares a thematic connection, our study concentrates on the implicit mechanisms through which over-parameterized models acquire the ability to discern and handle label noise. Consequently, our focus differs from their research.

In contrast to existing studies, our approach distinguishes itself by exploring the phenomenon through the lens of the learned feature space generated by trained models. The most related works to ours are the concurrent ones by Gamba et al. (2022), which measure sharpness at clean and noisily-labeled data points and empirically showed that smooth interpolation emerging both for large over-parameterized networks and trained large models confidently predicts the noisy training targets over large volumes around each training point. While their study also entailed an analysis involving separate evaluations of clean and noisily labelled data points, we employed the $k$-nearest neighbour algorithm to infer the inter-relationship between these two categories of training data in the learned feature space and demonstrated that for over-parameterized models, noisy labelled data tends to be classified alongside its genuine counterparts. Our approach suggests that the double descent phenomenon is a consequence of deep learning models learning how to separate the noise from the information in the training set.

## 3 METHODOLOGY

We first assess the contribution of data noise to the double descent phenomenon. Bartlett et al. (2020) describes an intuitive way of understanding properties of significant over-parameterization in linear regression: fitting the training data exactly but with near-optimal prediction accuracy occurs if and only if there are many low-variance (and hence, unimportant) directions in parameter space where the label noise can be hidden. While non-linear models are different from linear models in terms of parameter space, we propose that neural networks approach optimal prediction through a similar mechanism of concealing label noise. We hypothesize that the emergence of the double descent phenomenon can be attributed to the progressively over-parameterized models effectively isolating noisy data within the training set, thus diminishing the influence of interpolating these noisy data points.

Since this proposition is inspired by Nakkiran et al. (2021)'s empirical conclusion, we initiate our experimentation by replicating the double descent phenomenon as per the experimental framework outlined by this paper. This approach offers a comprehensively manageable setup and establishes

a comparative analysis of the phenomenon across diverse tasks and optimization techniques. With an increasing ratio of label noise introduced to the training dataset, the test error peak at the interpolation threshold becomes evident, signifying a definite correlation with the pronounced double descent phenomenon.

## 3.1 EXPERIMENT SETUP

We replicate the phenomenon of deep double descent across three frequently employed neural network architectures: Fully Connected NNs, Standard CNNs, and ResNet18, following the experiment setup of Nakkiran et al. (2021).

- **Fully Connected Neural Networks (FCNN):** We adopt a simple two-layer fully connected neural network with varying width $k$ on the first hidden layer, for $k$ in the range of [1, 1000]. The second fully connected layer is adopted as the classifier.
- **Standard Convolutional Networks (CNN):** We consider a family of standard convolutional neural networks formed by 4 convolutional stages of controlled base width $[k, 2k, 4k, 8k]$, for $k$ in the range of [1, 64], along with a fully connected layer as the classifier. The MaxPool is [2, 2, 2, 4]. For all the convolution layers, the kernel size = 3, stride = 1, and padding = 1. This architecture implementation is adopted from Nakkiran et al. (2021).
- **Residual Neural Networks (ResNet18):** We parameterize a family of ResNet18s from He et al. (2016) using 4 ResNet blocks, each consisting of two BatchNorm-ReLU-convolution layers. We scale the layer width (number of filters) of convolutional layers as $[k, 2k, 4k, 8k]$ for a varied range of $k$ within [1, 64] and the strides are [1, 2, 2, 2]. The implementation is adopted from `https://github.com/kuangliu/pytorch-cifar.git`.

We describe the experiment setup along with hyper-parameters we used in the training of all neural architectures below. All networks are trained using the Stochastic Gradient Descent (SGD) optimizer with zero momentum and Cross-entropy Loss for evaluation. A separate learning rate scheme is employed for different models: $lr = \frac{0.05}{\sqrt{1+[epoch/50]}}$ for FCNNs and $lr = \frac{0.05}{\sqrt{1+[epoch*10]}}$ for CNNs and ResNet18s. For every model, the learning rate starts with an initial value of 0.05 and updates every 50 epochs. A batch size of 128 is employed, and no explicit data augmentation or regularization techniques (including dropouts or pruning) are utilized in our experimental setup. Utilizing a sequence of models trained under a uniform training scheme showcases the double descent phenomenon across different models.

## 3.2 INTRODUCTION OF LABEL NOISE

While the presence of label errors has been established in the majority of commonly used benchmarking datasets including MNIST and CIFAR-10 (Zhang, 2017; Northcutt et al., 2021b), to facilitate comparative analysis of our findings, we intentionally introduce explicit label noise. In our experimental setup, label noise with probability $p$ signifies that during training, samples are assigned a uniformly random label with probability $p$, while possessing the correct label with probability $(1 - p)$ otherwise. We interpret a neural network with input variable $x \in R^d$ and learnable parameter $\theta$, incorporating all weights and biases. The training dataset is characterized as a collection of training points $(x_n, y_n)$, where $n$ ranges from 1 to $N$. A training dataset with $N$ data points comprises $m = p \times N$ noise labelled data points and $\hat{m} = (1 - p) \times N$ clean data points.

It is important to note that the label noise is sampled only once and not per epoch, and this original and random label information is stored before each experiment. Each experiment is replicated multiple times, and the final result is obtained by averaging the outcomes. Owing to the presence of inherent label errors, the probability $p$ is solely employed to signify the extent of explicit label noise and does not reflect the overall noise level present in the dataset. Our primary experimental conclusion is drawn from comparing results obtained under a dataset with a relatively noisy level.

## 3.3 LEARNED FEATURE SPACE INTERPRETATION STRATEGY

Our primary research inquiry aims to gain insights into the diverse strategies employed by deep neural networks of various widths in interpolating noisy labelled data in classification tasks. Based on

the existence of benign over-parameterization, we assume that test images, akin to training images mislabeled, are more likely to be correctly classified by over-parameterized models. For instance, we anticipate that an optimal classifier should yield accurate predictions for unseen images, even if it was trained on similar images with an adversarial label. Thus, we further hypothesize that the closest neighbours of mislabeled training images are correctly classified for a model with better generalization performance. To validate our hypothesis, we adopted the $k$-nearest neighbour algorithm with cosine similarity to interpret the relative locations of clean and noisy labelled data in the learned feature space. This methodology allows us to delineate the prediction strategy through an indirect approach and subsequently compare our findings with the overall generalization performance of these pre-trained models.

We consider L-layer neural networks, while representations on each layer are represented by $f_l(X)$, for $l = [1, ..., L]$. Our investigation centred on the penultimate layer representations before classification denoted as $f_{L-1}(X)$. Acknowledging the difficulties of interpreting high-dimensional feature spaces, our approach mainly entails characterizing the relative positioning of training data points. This involves establishing a connection between the hidden features of noisy data points in the learned feature space and their $k$-nearest neighbours within the complementary subset of clean data. So we calculate the prediction accuracy $P$ of mislabeled training data and the majority of its nearest neighbours in feature space are in the same class:

$$P = \frac{\sum_{i=1}^m [y_i = M(f_{L-1}(x_{i,1}), \ldots, f_{L-1}(x_{i,k}))]}{m}, \tag{1}$$

In this context, $(x_1, y_1), \ldots, (x_m, y_m)$ denotes the subset of noisy training data with original labels, and $(x_{i,1}, y_{i,1}), \ldots, (x_{i,\hat{m}}, y_{i,\hat{m}}))$ denotes the subset of clean training data as neighbours to $x_i$ for all $i = [1, \ldots, m]$. Given the cosine similarity $S_C(A, B) := \frac{A \cdot B}{\|A\| \|B\|}$ between two representations A and B, let $f_{L-1}(x_{i,1}), \ldots, f_{L-1}(x_{i,\hat{m}})$ be a reordering of the representations of the clean training data such that $S_C(f_{L-1}(x_i), f_{L-1}(x_{i,1})) \leq \cdots \leq S_C(f_{L-1}(x_i), f_{L-1}(x_{i,\hat{m}}))$. The closest $k$ neighbours are selected, and a Majority function $M(f_{L-1}(x_{i,1}), \ldots, f_{L-1}(x_{i,k}))$ describes the procedure for making predictions based on the majority label among the k-nearest neighbours. The predictions are subsequently compared with the original labels of noisy data, and $P$ denotes the proportion of accurate predictions among all the noisy data.

In line with our hypothesis, the prediction accuracy $P$ is indicative of the neural architecture's interpolation approach toward these noisy labelled data points. Due to the uncertain distribution of data and their representations, the $k$-NN method functions solely as an estimation technique, and our comparison focuses on variations in $P$.

In the next section, we present our empirical study and results of the learned feature space interpretation for trained neural networks of various widths in relationship to double descent.

## 4 EXPERIMENTS

In this section, we present our experiment results with a variety of neural architectures of increasing size trained for image classification tasks on either MNIST or CIFAR-10 datasets.

### 4.1 MNIST

We commence our exploration by revisiting one of the earliest experiments that revealed the presence of the double descent phenomenon, as presented in the work of Belkin et al. (2019). This experiment involved the training of a basic two-layer FCNN on a dataset comprising 4000 samples from the MNIST dataset ($N = 4000$). When zero label noise ($p = 0\%$) is introduced, as depicted in Figure.1(a), both the train and the test error curve decrease monotonically with increasing model width. The training error exhibits a faster decline and approaches zero at approximately $k = 15$, indicating the occurrence of an interpolation threshold under these conditions. The test error still decreases after this interpolation threshold, demonstrating a benign over-parameterization. This noticeable alteration in the shape of the test error curve at the interpolation threshold may be attributed to the intrinsic label errors of MNIST (Northcutt et al., 2021a).

With label noise ($p = 10\%/20\%$) introduced in the training dataset, a more distinguishable test error peak starts to form around the interpolation threshold as shown in Figure.1(b, c). Starting with small

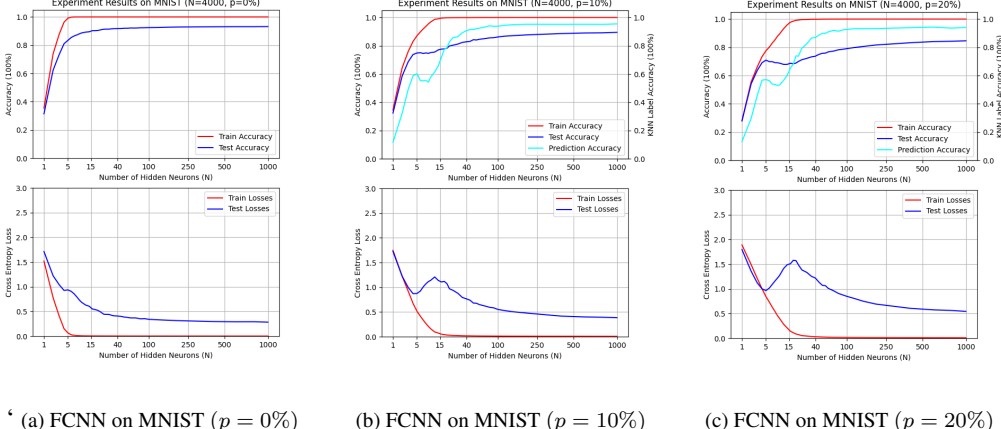

‘ (a) FCNN on MNIST ($p = 0\%$)   (b) FCNN on MNIST ($p = 10\%$)   (c) FCNN on MNIST ($p = 20\%$)

Figure 1: The phenomenon of double descent on two-layer FCNNs trained on MNIST ($N = 4000$), under varying explicit label noise ratios of $p = [0\%, 10\%, 20\%]$ and the prediction accuracy of noisy labelled data denoted as $P$ when $p > 0$. Optimized with SGD for 4000 epochs and decreasing learning rate. The test error curve of $p = [10\%, 20\%]$ performs the double descent phenomenon and the prediction accuracy $P$ of in-context noisy data learning showed a correlation generalization performance beyond context.

models, the test error curve exhibits a U-shaped pattern in the first half and the bias-variance trade-off is reached at the bottom of the U-shaped curve. The decrease of both train and test error before this bias-variance trade-off indicates the model undergoes an under-fitting stage. With the expansion of the model size, the model transitions into an overfitting regime, leading to a peak in the test error around the interpolation threshold, while the train error converges to 0. Finally, after surpassing this test error peak, the testing loss experiences a decline once more, showcasing how highly over-parameterized models can ultimately yield improved generalization performance. This intriguing behaviour is referred to as double descent. As the test loss undergoes a pattern of first decreasing, then increasing, and finally decreasing again, the test accuracy exhibits the opposite trend, i.e., first increasing, then decreasing, and then increasing once more. As Nakkiran et al. (2021) reported, the test loss peak is positively correlated with the label noise ratio $p$. In other words, as the label noise ratio increases, the peak in the test loss also tends to increase accordingly.

To comprehend the influence of noisy labelled data on the variation of the learned model during the double descent phenomenon, we examine the percentage $P$ of noisy data, and the majority of its nearest neighbours in feature space are in the same class. In this experiment setup, as depicted in Figure.1(b, c), the trajectory of the $P$ curve follows the pattern observed in the test accuracy curve, indicating a consistent alignment between the two. When test loss first decreases as the model expands, test accuracy and $P$ both rise. With the introduction of label noise, test accuracy decreases while $P$ fluctuates around 70%. When passing the interpolation threshold, both the test accuracy and $P$ significantly rise to 85% and nearly 100%, substantially affirming the validity of our hypothesis. In the meantime, the trend of test loss is opposite to both test accuracy and $P$, while changing correspondingly before and after the interpolation threshold. The computation of percentage $P$ relies solely on the training dataset and does not incorporate unseen data. Consequently, this percentage $P$ may be regarded as a weak predictor of generalization performance, though it necessitates knowledge of the distribution between clean and noisy data.

Recall that the noisy data points will be predicted based on their randomly assigned labels during this stage as the model fully interpolates the training set, we interpret the ascending $P$ curve in the over-parameterized region as a valid 'isolation' of the noisy labelled data in the learned feature space. When the neural network positions the learned representation of noisy data points closer to the class associated with their 'incorrect' labels, it results in lower $P$ prediction accuracy. Under these circumstances, similar test samples also tend to be predicted incorrectly due to the influence of these noisy labels. If the neural network successfully identifies the underlying data patterns and interpolates the correct samples around noisy data points, the prediction accuracy $P$ will be high by

nearby class members. Consequently, test samples resembling the noisy training data have a higher likelihood of being accurately identified due to their proximity to the interpolated surrounding clean data points and becoming less susceptible to the influence of the 'isolated' noisy labelled data. This explanation accounts for the improved generalization performance for over-parameterized models.

## 4.2 CIFAR-10

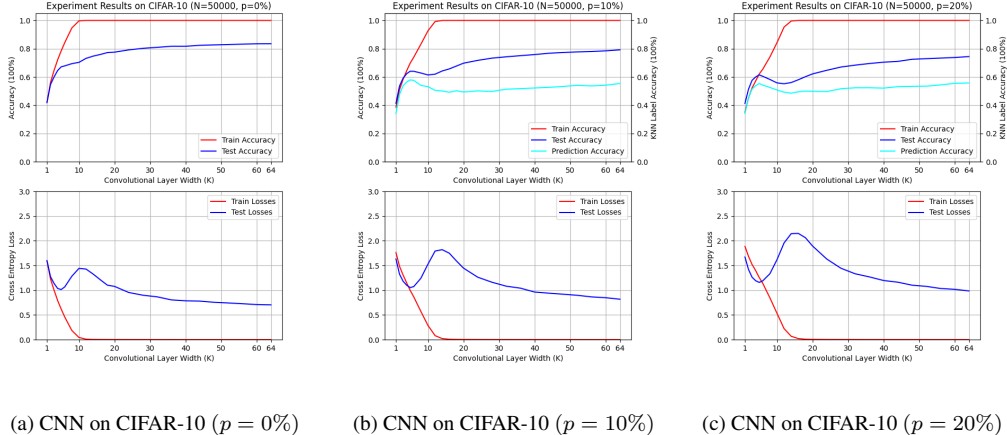

(a) CNN on CIFAR-10 ($p = 0\%$)  (b) CNN on CIFAR-10 ($p = 10\%$)  (c) CNN on CIFAR-10 ($p = 20\%$)

Figure 2: The phenomenon of double descent on five-layer CNNs trained on CIFAR-10 ($N = 50000$), under varying label noise ratios of $p = [0\%, 10\%, 20\%]$ and the prediction accuracy of noisy labelled data denoted as $P$ when $p > 0$. Optimized with SGD for 200 epochs and decreasing learning rate. The test error curve of $p = [0\%, 10\%, 20\%]$ performs the double descent phenomenon and the prediction accuracy $P$ of in-context noisy data learning showed a correlation generalization performance beyond context.

After a baseline case of deep learning with FCNNs on MNIST, we extended the experiment results to image classification on CIFAR-10, using two famous neural network architectures: CNN and ResNet. We first visit the experiment when the complete CIFAR-10 dataset ($N = 50000$) with zero explicit label noise ($p = 0\%$) is trained with five-layer CNNs: we observe that in Figure.4.2(a), a proposed double descent peak arises around the interpolation threshold. We posit that the variance in observations could stem from a higher ratio of label errors in CIFAR-10 compared to MNIST, as suggested by research estimates (Zhang, 2017; Northcutt et al., 2021b). Nevertheless, when additional label noise ($p = 10\%/20\%$) is introduced, the test error peak still increases accordingly, proving the positive correlation between noise and double descent according to Figure.4.2(b,c). While label noise increases the difficulties of models fitting the train set and shifts the interpolation threshold rightwards, the test error peak also shifts correspondingly.

When we review the noisy labelled data prediction test in the learned feature space, we can also observe an aligned correlation between the trend of generalization performance and prediction accuracy $P$. When test accuracy first increases and then decreases as the model expands, $P$ rises to about 60% and then decreases before the threshold when the peak of test error arises. Then, as the test error reduces and test accuracy rises, the $P$ curve rises again to about 60% for over-parameterized models. The observed correlation between the trend of test loss and $P$ remains consistent. The same conclusion can be derived from the experiment context Figure.3 of ResNet18s trained on the sample task. An important observation is that the prediction accuracy $P$ demonstrated in the learning experiments of CIFAR-10 is significantly lower compared to those on MNIST. We attribute this variance to the intricate feature space representations, which are more challenging to capture using cosine similarities. Our analysis focused more on how the interpolation strategy varied across the same neural architecture of different widths, and the k-NN methodology can only serve as an interpretation technique.

In conclusion, by constructing $P$ to represent the distribution of noisy labelled training data in the learned feature space, we propose that this k-NN prediction ability of the model is connected to the size of the model in the same way with double descent. Furthermore, We suggest that this correlation

can be attributed to the model's interpolation of correct samples around the noise labels, effectively isolating the misleading effect of the latter.

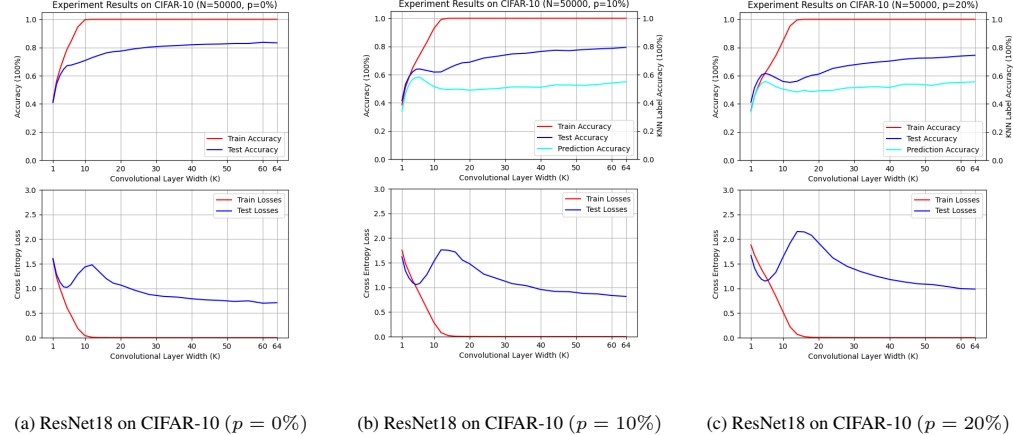

(a) ResNet18 on CIFAR-10 ($p = 0\%$)     (b) ResNet18 on CIFAR-10 ($p = 10\%$)     (c) ResNet18 on CIFAR-10 ($p = 20\%$)

Figure 3: The phenomenon of double descent on ResNet18s trained on CIFAR-10 ($N = 50000$), under varying label noise of $p = [0\%, 10\%, 20\%]$ and the prediction accuracy $P$ when $p > 0$. Optimized with SGD for 200 epochs and decreasing learning rate. The test error curve under $p = [0\%, 10\%, 20\%]$ performs the double descent phenomenon and the prediction accuracy $P$ of in-context noisy data learning showed a correlation to the generalization performance beyond context.

## 5 CONCLUSION

In this paper, we have studied the feature space of learned representations projected by various neural networks. By analyzing the behaviour of the model when achieving a perfect fit on the training set, we observe that: as the model's complexity and capacity increase, there is clear and distinctive isolation of noisy data points within their original 'class-mates' in the learned feature space. This intriguing observation indicates that broader neural networks interpolate correct samples throughout the vicinity of noisy samples in the learned feature space, effectively 'isolating' the impact of introduced label noise without explicit regularization techniques. We believe this implicit property plays a significant role in understanding how over-parameterization enhances generalization. We posit that the mechanism behind this phenomenon can be attributed to two key factors. First, the gradient descent optimization algorithm endeavours to strike a perfect balance among training data points to minimize the loss function, which can lead to the isolation of noisy data points. Second, the characteristics of high-dimensional space may facilitate a beneficial separation of noisy data points from clean data in the learned feature space.

From the perspective of this paper, the observed double decent phenomenon is associated with imperfect models, first behaving as predicted by the classical learning theory and increasing their error on the test set until they reach full interpolation on the train set. Then, they make use of the extra parameters above the interpolation point to better separate noise from information. We intentionally referred to 'imperfect models' because we claim that double descent should not be observable in perfect learners where perfect regularization should eliminate the effect. From our perspective, the double descent phenomenon is strictly related to imperfect models learning from noisy data.

Let's expand on this by restricting our considerations to supervised learning for classification problems: For an ideal training set with full information (no noise and no missing cases), a learner will learn the exact model at the interpolation point where the train set is exactly classified and the error becomes zero. In this case, at interpolation, also the test set will be exactly classified because the learner has learned the exact map. Before that interpolation, the error decreases with every new example. Clearly, this system does not show any double descent. At the interpolation point, the number of model parameters is approximately equal to the number of examples in the complete train set (i.e. the model is the matching table).

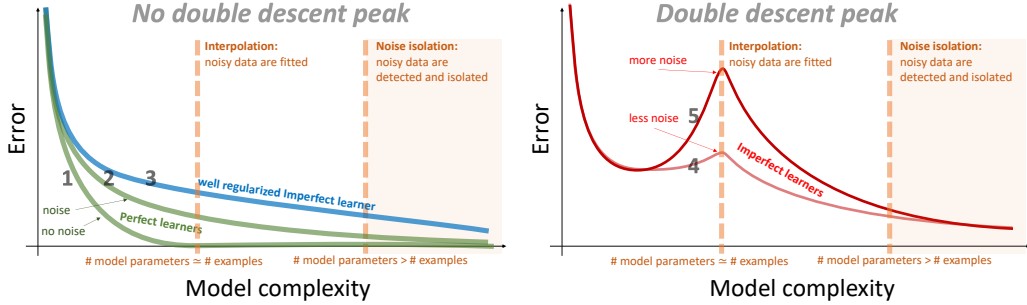

Figure 4: Exemplification of the emergence of the double descent effect in the perspective of the present paper. Starting from the bottom of the left panel: 1. the perfect learner, in the absence of noise, will learn the perfect model containing several parameters approximately equal to the number of significant examples; 2. in the presence of noise, the perfect learner needs to learn also to distinguish information from noise and it will learn a perfect model with several parameters which is higher the number of noisy examples; 3. the imperfect learner with optimal regularization will learn a well-performing model which distinguishes information from noise using several parameters which is higher the number of examples but should not show double descent peak. On the right panel: 4. the imperfect learner with sub-optimal regularization will learn a well-performing model first interpolating noise and signal and then learning to distinguish information from noise using several parameters which is higher the number of examples showing the double descent peak; 5. the larger is the noise, the higher the double descent peak is, and the larger is the number of parameters needed to distinguish information from noise.

In the presence of noise, the learner cannot learn the full map by interpolation. Indeed, the bias-variance trade-off predicts a U-shaped performance curve with first the learner increasing out-of-sample performances from digesting examples and then losing performances by trying to fit the in-sample noisy data. However, a learner who makes good use of regularization can do better and learn how to account for uncertainty. Likely, the number of model parameters needed to learn both the underlying map and to handle the noise is larger than the number of parameters needed to learn only the map in the ideal case without noise (the interpolation point). If the model is a perfect learner with perfect regularization, we do not expect the manifestation of any double descent, but rather a gradual improvement in out-of-sample performances with the number of parameters until a saturation point is reached where performances cannot be improved without further examples. We expect that this saturation point should occur at a number of parameters larger than the interpolation point. Double descent is instead observed for imperfect learners who at first try to interpolate the noise and then start an intrinsic regularization via over-parameterization. Our particular model of the emergence of the double descent phenomenon is illustrated In Fig. 4.

While the present study provides valuable insights into the mechanisms leading to the occurrence of the double descent phenomenon, a comprehensive theoretical framework is yet to be developed. Further investigations are needed to unravel the intricate details and offer a deeper understanding of this intriguing phenomenon. Another aspect that warrants attention is the impact of the architectures of deep neural networks. Studies have demonstrated that neural networks exhibit diverse behaviours based on their depth and width, even when the total number of parameters is identical. It piques our curiosity to investigate whether this variation in network shape could influence the underlying mechanisms of the double descent phenomenon. Another potential avenue for future exploration could involve implicit sparsity. We have observed that feature representations tend to exhibit sparsity in over-parameterized neural networks, particularly when dealing with high-dimensional feature spaces.

**Reproducibility Statement** To facilitate reproducibility, we provide a detailed description of the neural architectures and full training details including exact settings of hyper-parameters in Section 3. Our code is provided publicly available at `https://github.com/Yufei-Gu-451/double_descent_inference`.

## ACKNOWLEDGMENTS

The authors acknowledge the discussion with Siddharth Chaudhary, Joe Down, Nathan D'Souza, and Huijie Yan, all UCL students who took part in the academic year 2022-23 to project COMP0031. With them, we had invaluable discussions and great inspiration. This paper originated from that project and would not have been conceived without it. Please see Chaudhary et al. (2023) for the reference to that work. The author, T.A., acknowledges partial financial support from ESRC (ES/K002309/1), EPSRC (EP/P031730/1), and EC (H2020-ICT-2018-2 825215). X.Z. knowledges partial support from the National Natural Science Foundation of China (No. 62076068).

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

# A  EXPERIMENT DETAILS

## A.1  SCALING OF MODEL PARAMETERIZATION

The scaling of model parameterization to the universal layer width unit $K$ is shown in Figure.5. In our experiment results, we labelled the x-axis scale as the model width parameter $k$ instead of the scaling of parameterization. However, we believe this labelling does not impact our conclusion because of the monotonic relationship between these two factors.

## A.2  MODEL PERFORMANCE IN THE PRESENCE OF HIGH LABEL NOISE

Here, we provided two additional experiment results when a high label noise ratio $p = 40\%$ is introduced to the training dataset of MNIST and CIFAR-10 that, due to page limitations, could not be included in the main context. Noise has a substantial impact on the model's classification accuracy, leading to a reduction in test accuracy. It is evident that the prediction accuracy curve $P$ exhibits significant instability when subjected to a high percentage of noisy samples, with correct samples becoming sparser in the learned feature space. However, the correlation between $P$ and generalization performance remains consistent.

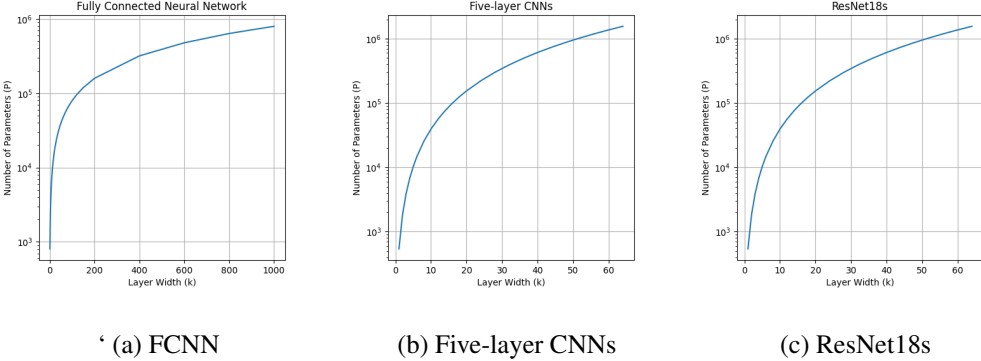

'(a) FCNN                (b) Five-layer CNNs                (c) ResNet18s

Figure 5: Scaling of the number of parameters as model size with layer width unit k of the three neural architectures used in our experiments in the Methodology section **??**. We apply a logarithmic scale to the parameter counts $P$ of all neural architectures.

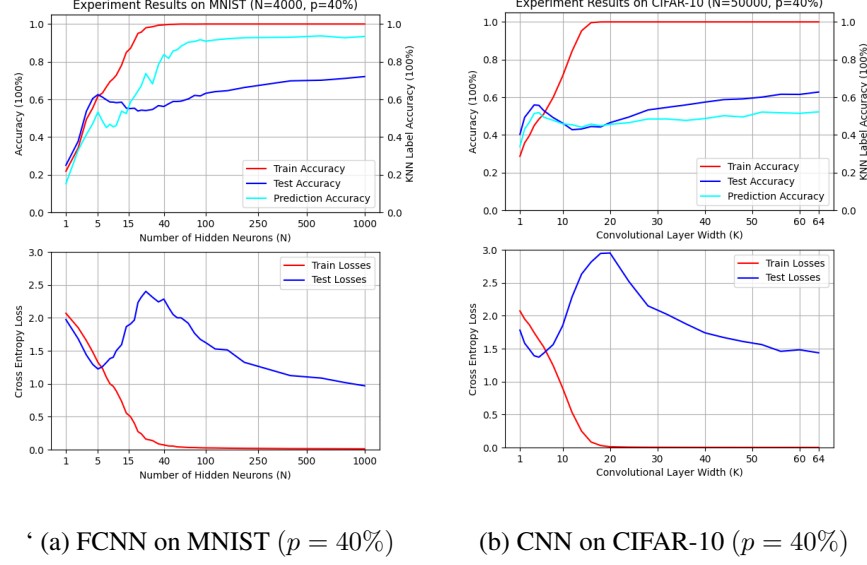

'(a) FCNN on MNIST $(p = 40\%)$          (b) CNN on CIFAR-10 $(p = 40\%)$

Figure 6: The phenomenon of double descent and the prediction accuracy $P$ when a high label noise of $p = 40\%$ is introduced on the training dataset.

