# OpenReview forum: "Unraveling the Enigma of Double Descent: An In-depth Analysis through the Lens of Learned Feature Space"
_ICLR.cc/2024/Conference — ICLR 2024 poster_

### Official Review · Reviewer_Ea52 · 2023-10-26

**Soundness:** 2 fair
**Presentation:** 2 fair
**Contribution:** 1 poor
**Rating:** 5
**Confidence:** 3

**Summary:**

This paper presents an empirical study of the relation between model capacity and generalization, while considering the learned feature space. In particular, it analyzes the phenomena of double descent, in which test performance improves after a certain interpolation threshold, contrary to the traditional U-shaped curve.  This work argues that models with small capacity first overfit to the noise present in the data and that overparameterized models then learn to “recognize” noisy labels, separating signal from noise.

**Strengths:**

In contrast to most previous empirical works on double descent, which only analyze cross entropy loss, double descent in terms of accuracy is also studied. As shown in their plots (see Figures 1, 3 and 5) the double descent peak is considerably smaller (sometimes even absent) when looking at accuracy. tSNE visualizations give more intuition about the phenomenon, and experiments are performed for 3 models and 3 noise levels and 2 datasets.

**Weaknesses:**

- One main weakness is the fact that a big part of the paper is devoted to reproducing experiments from [1]. In these experiments, the only significant difference is the fact that they report both accuracy and cross entropy loss.

- I fully agree with the author’s comment on section 3 regarding tSNE maps: “While this visualization may not accurately represent the intricate inherent structure of high-dimensional features, it aims to gain insights into the clustering and distribution of data points, thereby enhancing our understanding of the model’s internal representations.” While tSNE maps might give intuition about a phenomenon, they can also fabricate patterns and might not be sufficient evidence for a proposition.

- The writing is sometimes confusing. For instance, in section 3.3: “we calculate the prediction accuracy Kp of noisy labelled data of the original clean labels for each noisy labeled data point matching the label prediction of its k-nearest neighbors in the learned feature space”.

 Thus, I do not see substantial empirical nor theoretical contributions in this work.

[1] DEEP DOUBLE DESCENT: WHERE BIGGER MODELS AND MORE DATA HURT Nakkiran et al.

**Questions:**

-

---

> ### Author Response · Authors · 2023-11-19
> **Response to official review**
>
> Firstly, we appreciate your time in reviewing our paper and offering valuable comments and insights.
>
> Q1: ``One main weakness is the fact that a big part of the paper is devoted to reproducing experiments from [1]. In these experiments, the only significant difference is the fact that they report both accuracy and cross-entropy loss."
>
> A1: We did replicate the experiment setups from [1] as we stated in our Experiment Setup section. Nonetheless, our primary focus in research is on the K-NN prediction test carried out on data with noisy labels. Our conclusions are drawn through a comparative analysis involving the interplay between the double descent phenomenon and prediction accuracy across varying levels of introduced label noise. Embracing a foundational and validated experimental setup proves advantageous for facilitating straightforward comparisons and validating our findings.
>
> Q2: ``While tSNE maps might give intuition about a phenomenon, they can also fabricate patterns and might not be sufficient evidence for a proposition."
>
> A2: We agree that the t-SNE experiments lack persuasiveness and have only a loose connection to our primary hypothesis. We have now reallocated the experiments to the appendix.
>
> Q3: ``The writing is sometimes confusing. For instance, in section 3.3: “we calculate the prediction accuracy Kp of noisy labeled data of the original clean labels for each noisy labeled data point matching the label prediction of its k-nearest neighbors in the learned feature space”."
>
> A3: We have also strengthened the descriptions of our methodology, which includes a detailed explanation of how our hypothesis is founded, why our experiment can be used to validate our hypothesis, and re-formulated our mathematical description.
>
> Q4: ``I do not see substantial empirical nor theoretical contributions in this work."
>
> A4: Our distinctive contribution lies in illuminating the influence of model parameters on the strategy and effectiveness of learning representations. Furthermore, our study unveils the process by which over-parameterized models discern and effectively "isolate" label noise within the feature space. These are previously undocumented phenomena that have not been extensively explored or discussed in the existing body of literature.

---

> > ### Comment · Reviewer_Ea52 · 2023-11-21
> >
> > Thanks for addressing some of my concerns.
> >
> > It is now clearer that the primary goal is to analyze the K-NN prediction test, and perform a comparison with the experiments from 1. However, I still hold the view that the experiments carried out do not provide sufficient evidence for a solid hypothesis as to why over-parameterized models isolate label noise.

---

> > > ### Author Response · Authors · 2023-11-21
> > > **Response to concerns**
> > >
> > > The capacity of over-parameterized models to discern and segregate noise within the learned feature space is
> > > a phenomenon we unveiled through the k-NN prediction test. We further showcased that this phenomenon
> > > exhibits a profound correlation with double descent and may serve as its underlying internal mechanism.
> > > Exploring the underlying reasons for this phenomenon is not the primary focus of this paper, and we look
> > > forward to delving into it more thoroughly in future analyses. Nevertheless, we greatly value your response.

---

### Official Review · Reviewer_d6tR · 2023-10-27

**Soundness:** 2 fair
**Presentation:** 2 fair
**Contribution:** 2 fair
**Rating:** 5
**Confidence:** 5

**Summary:**

The research aims to show that the phenomenon of double descent with respect to model size appears in neural networks due to the addition of noisy labels, i.e., smaller models learn this noise, while larger size models are able to avoid it and recognize wrong labels while learning useful information. The proposed approach to this aim is to measure the nearest neighbor prediction for mislabeled examples based on learned representations (on the penultimate layer) and then demonstrate that this prediction corresponds to the original (true) label, no matter that network "correctly" predicts wrong label. t-SNE visualizations of representations are used as an additional evidence to this point.

**Strengths:**

Investigation of the ability of large neural networks to interpolate noisy labels, but nevertheless have good generalization is an important contribution to the current research. The mechanisms behind this phenomenon shed light on the generalization abilities of such models.

**Weaknesses:**

The abstract of the paper states that the main contribution is to demonstrate that model-size double descent is happening only with label noise present in the dataset. Nevertheless, the experiment with CIFAR100in the paper directly disproves this claim already. Further, it is claimed that double descent will not happen in the correctly regularized networks, but no regularization is ever discussed in the paper itself, except for an unusual claim that overparameterization is a form of regularization (which is also never discussed in the paper). Together, this leaves only one possible contribution - explaining how interpolating networks deal with mislabeled examples. This is aimed to be explained through nearest neighbors classification on the representations for mislabeled examples - showing that this accuracy is very high for interpolating models. Nevertheless, only the MNIST experiment indeed demonstrates high accuracy of these predictions together with the high accuracy on noisy task, in other experiments this accuracy is considerably low. t-SNE visualizations can be simply explained by the ability of larger models to get a more distinguished representations for different classes, but the positioning of the noisy labeled ones does not change much (see class 0 in fig.2 for example). Thus, the last contribution is not significantly supported by the results.

Minor:

- please use \citep and \citet for the citations outside of the sentence and in the sentence

- please check the grammar and typos in the paper

- please improve the explanation of the formula (1), it is currently extremely convoluted

- the font of the title does not correspond to ICLR style

**Questions:**

1 - What is the main goal of the research done?

2 - What is the observation made with respect to the selected metric (nearest neighbors) from the double descent plots?

---

> ### Author Response · Authors · 2023-11-19
> **Response to official review**
>
> Firstly, we appreciate your time in reviewing our paper and offering valuable comments and insights.
>
> Q1: ``The abstract of the paper states that the main contribution is to demonstrate that model-size double descent is happening only with label noise present in the dataset. Nevertheless, the experiment with CIFAR100in the paper directly disproves this claim already."
>
> A1: In the abstract, we stated that "its occurrence is strongly influenced by the presence of noisy data.". Speaking of the CIFAR-10 experiments, "We suppose that this phenomenon may arise from the intrinsic noise and increased complexity associated with image recognition of CIFAR-10 in contrast to MNIST.". I believe we have already addressed and explained your first concern in our paper.
>
> Q2: "It is claimed that double descent will not happen in the correctly regularized networks, but no regularization is ever discussed in the paper itself, except for an unusual claim that overparameterization is a form of regularization (which is also never discussed in the paper)."
>
> A2: We reach the conclusion that double descent should not occur in well-regularized models from referencing previous research and a derivation of our analysis of imperfect learners and noisy data. This conclusion line is now removed from our paper.
>
> Q3: ``Nevertheless, only the MNIST experiment indeed demonstrates high accuracy of these predictions together with the high accuracy on noisy tasks, in other experiments this accuracy is considerably low."
>
> A3: Our explanation of how interpolating networks deal with mislabeled examples is demonstrated through the aligned trend of test accuracy and the percentage ${\rm P}$ (previously denoted as ${\rm Kp}$), instead of a high value. Although the accuracy of predictions on CIFAR-10 is relatively modest, it's worth noting that the performance of CNNs and ResNets on this dataset is also suboptimal. Considering the challenge of a 10-class classification problem, achieving nearly 60\% accuracy is commendable, and the comparison between models of varying widths still yields conclusive results.
>
> Q4: ``t-SNE visualizations can be simply explained by the ability of larger models to get more distinguished representations for different classes, but the positioning of the noisy labeled ones does not change much (see class 0 in fig.2 for example)."
>
> A4: We acknowledge that the t-SNE experiments lack persuasiveness and have only a loose connection to our primary hypothesis. We have now reallocated the experiments to the appendix.
>
> Q5: ``What is the main goal of the research done?"
>
> A5: Our distinctive contribution lies in illuminating the influence of model parameters on the strategy and effectiveness of learning representations. Furthermore, our study unveils the process by which over-parameterized models discern and effectively "isolate" label noise within the feature space. These are previously undocumented phenomena that have not been extensively explored or discussed in the existing body of literature.
>
> Q6: ``What is the observation made with respect to the selected metric (nearest neighbors) from the double descent plots?"
>
> A6: In all of our experiments, the trajectory of the ${\rm P}$ curve (denoted as ${\rm Kp}$ previously) is in accordance with the pattern observed in the test accuracy curve, indicating a consistent alignment between the two. This alignment exists both before and after the interpolation threshold. This observation underscores a statistical correlation between the local structures within the learned representations of noisy training data and the overall performance in generalization. This validation serves to affirm the accuracy of our hypothesis.
>
> Minor corrections regarding grammar, typos, and formatting of titles and citations have been adopted. Equation 1 has been reformulated to enhance clarity and understanding.

---

> > ### Comment · Reviewer_d6tR · 2023-11-20
> >
> > I thank the authors for the answers.
> >
> > A1. I am sorry for a typo with the dataset name, it should be CIFAR-10 and not CIFAR-100. Nevertheless, my point stands. If you claim that double descent is influenced by noise, then argument about intrinsic noise is vague and not verifiable. Please try to demonstrate then that CIFAR-10 indeed has noise inside of itself as compared to MNIST that does not.
> >
> > A3. I am not questioning the accuracy of the model used for CIFAR-10, but the difference between test accuracy of the model and the accuracy of knn predictions. In MNIST experiments the accuracy of knn is even higher than the original one, while for CIFAR-10 it is significantly lower. How would you interpret this? Is only the trend in the development enough to make the conclusions you make?
> >
> > A5. How exactly do you illuminate "the influence of model parameters on the strategy and effectiveness of learning representations"?
> >
> > A6. The alignment of trajectories can be overall explained by a better generalization ability of the model as well.
> >
> > With the new explanation given to the calculation of the metric, I have an additional question: you check the alignment of predictions using the mislabeled training sample and the neighbors among the _test set_. My guess is that this result cannot be saying: we do expect the network to be consistent in its predictions. Therefore indeed all the similar inputs will get same label. I think it might be much more interesting how interpolating networks interpolate the training data, do they indeed learn features similarly even for the mislabeled samples.

---

> > > ### Author Response · Authors · 2023-11-21
> > > **Reply to questions and discussions**
> > >
> > > Thank you for your further comments and suggestions.
> > >
> > > Regarding A1, pervasive label noise has been present in commonly utilized datasets, with validation studies revealing higher (label) error rates in CIFAR-10 compared to MNIST~[2]. Furthermore, upon closer examination of the double descent plot for MNIST trained without label noise, a noticeable alteration in the shape of the test error curve is observed at the interpolation threshold — a discernible rise. We contend that it is not the case that double descent occurs on CIFAR-10 but not on MNIST; rather, the distinction lies in the level of the interpolation peak, and this difference is due to the different label error rates.
> > >
> > > Speaking of A3, the trend in development is enough to make the conclusions and validate our hypothesis. Drawing a comparison between the actual test accuracy and prediction accuracy (expressed as a percentage ${\rm P}$) is meaningless when label noise is absent, as the context for a k-NN comparison becomes non-existent. Our objective is to discern the correlation between the enhanced double descent phenomenon and the model's interpolation strategy concerning introducing label noise when explicit label noise is introduced.
> > >
> > > For A5/A6, the better generalization ability of over-parameterized models has been documented through empirical studies showcasing the occurrence of the double descent phenomenon. The primary objective of this study is to unravel the underlying mechanisms driving this phenomenon, particularly exploring its association with label noise in the form of noisy labeled data. We demonstrated that over-parameterized models, when contrasted with under-parameterized ones, exhibit a more precise probability of predicting correctly in the vicinity of mislabeled data points within the training set; While the exact training point are still predicted with the incorrect label by these fully-interpolate models.
> > > We posit that this phenomenon has not been previously documented and could serve to elucidate the internal mechanism of the double descent phenomenon.
> > >
> > > Addressing your last inquiry, it is indeed a valid question whether we should compare the noisy labeled data to its counterparts in the training set or the test set. We contend that this choice does not markedly impact the validity of our hypothesis because the neighbors only serve as a convenient means of characterizing local structures in the high-dimensional space. As supporting evidence, we conducted empirical studies in both scenarios (initial submission on the training set; and revision on the test set), and the observations consistently align. Nevertheless, the prediction accuracy on the training set typically exceeds that on the testing set, often reaching full interpolation, and how interpolating networks interpolate the training data is surely more interesting. We have reverted to the initial submission version, incorporating experiment results comparing the training set, and modified the methodology description.
> > >
> > > [2] Northcutt, C.G., Athalye, A. and Mueller, J., 2021. Pervasive label errors in test sets destabilize machine learning benchmarks. arXiv preprint arXiv:2103.14749.

---

> > > > ### Comment · Reviewer_d6tR · 2023-11-22
> > > >
> > > > I thank the authors for further clarifications.
> > > >
> > > > A1. Please add this citation to your paper. Ideally, it would be nice to then position CIFAR-10 as one with default label noise level (researched in the paper you provided) and as 0% level, because in the latter case your conjectures look weaker.
> > > >
> > > > A3. My claim is the following: is the model indeed interpolates around all the noisy (in the training set) samples in the correct way, then k-nn should perform as nice as the model itself. I agree nevertheless, that the trend still shows that this k-nn ability of the model is connected to the size of the model in the same way with double descent. I think this should be stated more clearly in the paper.
> > > >
> > > > I think it is great that you also look into the training set interpolation.
> > > >
> > > > Overall, I do change my opinion and I agree with the authors that this investigation can be interesting. I would suggest putting different emphasis on the overall description and positioning it as an interpolating models investigation. I also encourage the authors to make more precise mathematical descriptions (in the renewed one you still have some sloppy notations, for example, naming sum of labels as a set of samples). I will therefore raise my score.

---

> > > > > ### Author Response · Authors · 2023-11-22
> > > > >
> > > > > For A3, as the exact distribution of data and their representations is uncertain, the k-NN method can only serve as an estimation technique. For instance, the k-NN prediction may perform inadequately on outliers that lack nearby neighbors. The distribution of neighbors surrounding each noisy labeled data point will also influence the outcome, thereby impacting the neural network's interpolation result. The exact reason behind the different circumstances required further investigation. We have stated the property of the k-NN prediction more clearly in the Methodology section.
> > > > >
> > > > > Besides this, we have also:
> > > > > 1. Rephrased the claim and conclusions of our paper in the Introduction, the Methodology, and the Conclusion section emphasizing the interpolating model's investigation. We hope we have cited your words properly. We are also willing to rephrase the abstract if permitted.
> > > > > 2. Further enhanced our mathematical descriptions with the methodology.
> > > > > 3. Added citations of research studying label errors in benchmarking datasets and discussions in the Methodology and Experiments section. (We didn't configure the graph notations, since the label error rates are estimates instead of authorized claims. We refer $p$ to the explicit ratio of label noise that we introduced.)
> > > > >
> > > > > We sincerely thank you for your suggestions and advice.

---

### Official Review · Reviewer_9d7L · 2023-10-30

**Soundness:** 2 fair
**Presentation:** 2 fair
**Contribution:** 3 good
**Rating:** 3
**Confidence:** 3

**Summary:**

This article focuses on "Double Descent" phenomenon, suggesting that the ability of a model to adapt to and recognize noisy data has a significant impact on its overall generalization performance and contributes to the phenomenon of double descent, and proposing the use of a k-nearest neighbor algorithm to infer the relative positions of clean and noisily labeled data in the learned feature space.

**Strengths:**

1.This paper is feasible to understand the generalization ability and double descent phenomenon through the perspective of noise.

**Weaknesses:**

1.The methodology section of this paper (only 14 lines of text and only 6 lines of description of the actual methodology) is poorly described and lacks math-related descriptions, making it difficult to accurately understand the author's intention.

2.Is it optimal to use K-nearest neighbors here? the feature space is probably in a high-dimensional Manifolds, have the authors considered more complex cases?

3. I don't think this article convinces me enough about the mechanism by which it reveals that over-parameterization enhances generalization. The mechanism between over-parameterization and generalization cannot be revealed in depth only by the loss curves and T-sne images under a small number of experiments.

4. It is well recognized that over-parameterization is beneficial, especially since recent large models have similarly benefited from over-parameterization, and the authors should discuss the advantages of this paper in the context of recent over-parameterization research.

**Questions:**

See weakness.

---

> ### Author Response · Authors · 2023-11-19
> **Response to official reviews**
>
> Firstly, we appreciate your time in reviewing our paper and offering valuable comments and insights.
>
> Q1: `` The methodology section of this paper is poorly described and lacks math-related descriptions, making it difficult to accurately understand the author's intention."
>
> A1: Thank you for your kind advice, we have now improved the methodology with a more complete description of how our hypothesis is founded, why our experiment can be used to validate our hypothesis, and re-formulated our mathematical description. Some paragraph is cited below:
> ``Based on the existence of benign over-parameterization, we assume that test images, akin to training images mislabeled, are more likely to be correctly classified by over-parameterized models. For instance, we anticipate that an optimal classifier should yield accurate predictions for unseen images, even if it was trained on similar images with an adversarial label. Thus, we further hypothesize that the closest neighbours of mislabeled training images are correctly classified. ... We calculate the percentage ${\rm P}$ of mislabeled training data, and the majority of its nearest neighbours in feature space are in the same class ..."
>
> Q2: ``Is it optimal to use K-nearest neighbors here?"
>
> A2: The application of a K-nearest neighbors (KNN) algorithm in a high-dimensional feature space can be reasoned based on several considerations. In a high-dimensional feature space, the notion of proximity becomes more nuanced, and traditional distance measures may lose their discriminatory power. KNN, as a non-parametric and instance-based algorithm, relies on the concept of similarity between data points (we use cosine similarity as our measurement tool). While we are trying to infer and observe local structures, KNN remains an effective tool for exploratory analysis.
>
> Q3: ``The mechanism between over-parameterization and generalization cannot be revealed in depth only by the loss curves and T-sne images under a small number of experiments."
>
> A3: Our explanation of how interpolating networks deal with mislabeled examples is demonstrated through the trajectory of the P curve (denoted as Kp previously) in accordance with the pattern observed in the test accuracy curve, indicating a consistent alignment between the two. This alignment exists both before and after the interpolation threshold. This observation underscores a statistical correlation between the local structures within the learned representations of noisy training data and the overall performance in generalization. This validation serves to affirm the accuracy of our hypothesis.
>
> Q4: ``The authors should discuss the advantages of this paper in the context of recent over-parameterization research."
>
> A4: We have added discussions of recent over-parameterization research in the related works section. Since we haven't encountered comparable hypotheses in previous literature, our distinctive contribution within the realm of over-parameterization research lies in illuminating the influence of model parameters on the strategy and effectiveness of learning representations. Furthermore, our study unveils the process by which over-parameterized models discern and effectively "isolate" label noise within the feature space. These are previously undocumented phenomena that have not been extensively explored or discussed in the existing body of literature.

---

### Official Review · Reviewer_P23K · 2023-10-30

**Soundness:** 3 good
**Presentation:** 3 good
**Contribution:** 2 fair
**Rating:** 5
**Confidence:** 2

**Summary:**

The paper performs a series of empirical studies into the double descent phenomena. The main take-away is that double descent appears when the model is faced with label noise.

**Strengths:**

(I stress that I have low confidence in my review. It is highly likely that I missed something important)

* The double descent is, currently, highly counter-intuitive to many, so any insights into the phenomena are valuable.
* The stated hypothesis regarding label noise is clear and seems intuitive.

**Weaknesses:**

(I stress that I have low confidence in my review. It is highly likely that I missed something important)

* As a non-expert, I interpret the experiments as showing that double descent phenomena appear with label noise in large models. I acknowledge that this is an interesting observation, but it does not seem sufficient to justify conclusions such as double descent not appearing in well-regularized models. I can see the intuition, but, in my non-expert view, the experimental data seems insufficient.
* The paper considers only a small selection of data sets and models. It is not clear to me if such is sufficient to draw conclusions.
* The paper does not provide a theoretical explanation (which is fine, but then I had wished for a wider selection of experiments).
* I found it very difficult to read the t-SNE plots.
* [minor] It appears that the ICLR formatting instructions were disregarded when changing the font size of the paper title.

**Questions:**

* What is the 'wedge product' in Eq. 1?

---

> ### Author Response · Authors · 2023-11-19
> **Response to official review**
>
> Firstly, we appreciate your time in reviewing our paper and offering valuable comments and insights.
>
> Q1: ``It does not seem sufficient to justify conclusions such as double descent not appearing in well-regularized models."
>
> A1: We reach the conclusion that double descent should not occur in well-regularized models from referencing previous research and a derivation of our analysis of imperfect learners and noisy data. This conclusion line is now removed from our paper.
>
> Q2: ``The paper considers only a small selection of data sets and models. It is not clear to me if such is sufficient to draw conclusions."
>
> Secondly, the paper delves into a limited array of datasets and models. This deliberate selection is based on the recognition that larger and more intricate datasets inherently introduce additional noise, potentially interfering with the precision of our experiments. Our focus has been on conducting empirical studies within foundational experiment setups in deep learning. We assert that these chosen scenarios not only align with widely accepted practices but also serve as fundamental benchmarks for validating deep learning techniques.
>
> Q3: ``The paper does not provide a theoretical explanation (which is fine, but then I had wished for a wider selection of experiments)."'
>
> A3: We contend that the empirical studies conducted in our research encompass various fundamental experiment setups in deep learning, specifically addressing real-world classification problems. Artificial data may lack sufficient persuasiveness, as the intricacy and noise inherent in more challenging tasks, as well as state-of-the-art methods, can significantly impact the actual structures of the feature space, introducing numerous uncontrollable factors. Our distinctive contribution lies in illuminating the influence of model parameters on the strategy and effectiveness of learning representations. Furthermore, our study unveils the process by which over-parameterized models discern and effectively "isolate" label noise within the feature space. These are previously undocumented phenomena that have not been extensively explored or discussed in the existing body of literature.
>
> Q4: ``I found it very difficult to read the t-SNE plots."
>
> A4: We acknowledge that the t-SNE experiments lack persuasiveness and have only a loose connection to our primary hypothesis. We have now reallocated the experiments to the appendix.
>
> Minor corrections regarding grammar, typos, and formatting of titles and citations have been adopted. Equation 1 has been reformulated to enhance clarity and understanding.

---

### Author Response · Authors · 2023-11-22
**Appreciation to all Reviewers**

To All Reviewers:

We would like to thank the reviewers for their insightful comments and efforts put into providing high-quality reviews. We plan to release the source codes of our experiments to the research community. We appreciate all of you for your comments highlighting the strengths of our work for a summary.

During the review stage, we have:

- rephrased the findings and conclusions of our study in the introduction and the conclusion section

- including a discussion on recent studies regarding benign over-parameterization

- strengthened our explanation of the hypothesis and proposed methodology, with a reformulated formula

- refined our analysis of the experiment results

- performed minor corrections for grammar, typos, and formatting of the title and citations and enhanced our written content

- reallocated t-SNE experiments to the appendix

We express our sincere gratitude to the reviewers for their valuable feedback and constructive questions aimed at enhancing the quality of our manuscript. We have thoroughly addressed the raised queries. If our responses satisfactorily address your concerns and you consider our paper acceptable, could you kindly consider adjusting the scores accordingly? We greatly appreciate your consideration. Thank you in advance! :-D

---

### Meta-Review · Area_Chair_jYcu · 2023-12-13

**Metareview:**

This paper tries to understand why double descent occurs in over-parameterized networks. The authors hypothesise that model-size double descent occurs because of the label noise. Authors show through standard experiments that the models could try to fit the noise in the label and then reorganize and learn to separate the noise from the signal.

There were several points raised by the reviewers and the authors addressed most of the concerns. It is worth noting that 2 reviewers changed their scores after the discussions. Reviewer 9d7L gave a score of 3 but I do not see any valid criticism that is not addressed by the authors.

Given that the paper provides an interesting explanation for why double descent happens and there is no major issue with the claims in the paper, I think the paper is worth publishing.

**Justification For Why Not Higher Score:**

This paper has some interesting explanations on why double descent happens. But there is no strong theory for it. Having no theory is ok. but then, I would expect way more experiments.

**Justification For Why Not Lower Score:**

There is no reason to reject the paper. I do not want to reject it just because their experiments are only on MNIST and CIFAR.

---

### Decision · Program_Chairs · 2024-01-16

Accept (poster)